# Apitoxin and Its Components against Cancer, Neurodegeneration and Rheumatoid Arthritis: Limitations and Possibilities

**DOI:** 10.3390/toxins12020066

**Published:** 2020-01-21

**Authors:** Andreas Aufschnaiter, Verena Kohler, Shaden Khalifa, Aida Abd El-Wahed, Ming Du, Hesham El-Seedi, Sabrina Büttner

**Affiliations:** 1Department of Biochemistry and Biophysics, Stockholm University, Svante Arrheniusväg 16, 106 91 Stockholm, Sweden; andreas.aufschnaiter@su.se; 2Department of Molecular Biosciences, The Wenner-Gren Institute, Stockholm University, Svante Arrheniusväg 20C, 106 91 Stockholm, Sweden; verena.kohler@su.se (V.K.); shaden.khalifa.2014@gmail.com (S.K.); 3Department of Bee Research, Plant Protection Research Institute, Agricultural Research Centre, 12627 Giza, Egypt; aidia_2006@yahoo.com; 4Pharmacognosy Group, Department of Medicinal Chemistry, Uppsala University, Biomedical Centre, Box 574, 751 23 Uppsala, Sweden; 5Department of Chemistry, Faculty of Science, Menoufia University, 32512 Shebin El-Kom, Egypt; 6School of Food Science and Technology, National Engineering Research Center of Seafood, Dalian Polytechnic University, Dalian 116024, China; duming@dlpu.edu.cn; 7International Research Center for Food nutrition and safety, Jiangsu University, Zhenjiang 212013, China; 8Institute of Molecular Biosciences, University of Graz, Humboldtstraße 50, 8010 Graz, Austria

**Keywords:** apamin, apitoxin, bee venom, cancer, melittin, neurodegeneration, phospholipase A2, rheumatoid arthritis

## Abstract

Natural products represent important sources for the discovery and design of novel drugs. Bee venom and its isolated components have been intensively studied with respect to their potential to counteract or ameliorate diverse human diseases. Despite extensive research and significant advances in recent years, multifactorial diseases such as cancer, rheumatoid arthritis and neurodegenerative diseases remain major healthcare issues at present. Although pure bee venom, apitoxin, is mostly described to mediate anti-inflammatory, anti-arthritic and neuroprotective effects, its primary component melittin may represent an anticancer therapeutic. In this review, we approach the possibilities and limitations of apitoxin and its components in the treatment of these multifactorial diseases. We further discuss the observed unspecific cytotoxicity of melittin that strongly restricts its therapeutic use and review interesting possibilities of a beneficial use by selectively targeting melittin to cancer cells.

## 1. Introduction

Natural products represent important and extensively used sources for the discovery of novel therapeutics. Compounds isolated from plants, animals and microorganisms are successfully applied in modern medicine, and a wide range of these natural substances exhibit antimicrobial and anticancer properties, as well as neuroprotective characteristics [1]. Furthermore, plant isolates employed in traditional Chinese medicine show anticancer and anti-inflammatory features in preclinical studies [2]. Based on these findings, detailed characterization of purified components from these compounds and comprehensive analysis of potential pharmacological effects might contribute to the development of novel therapies. Even though it seems counterintuitive at first glance, several animal venoms, including apitoxin (honey bee venom), have beneficial effects against various diseases. Bee venom is produced in a gland in the abdominal cavity of honey bees (e.g., *Apis mellifera*) and is a complex mixture of biologically active peptides, including melittin or apamin; mast cell degranulating peptide; adolapin; amines such as histamine, dopamine and noradrenalin; enzymes including phospholipase A2 and B; hyaluronidase; diverse carbohydrates; and lipids [3] (please see Table 1 for a list of the major bee venom components). Biological characterization of this venom revealed that many of these peptides target ion channels and receptors of the peripheral and central nervous system [4].

Allergies against insect venoms are frequently observed, including anaphylactic reactions with potential fatal outcomes [6]. Major allergens in bee venom include phospholipase A2, melittin and hyaluronidase [6,7], which can lead to a variety of symptoms including mild (skin reactions, flushing, urticaria and angioedema), moderate (dizziness, dyspnea and nausea) and even severe reactions like anaphylactic shock, loss of consciousness and cardiac or respiratory arrest often accompanied with high mortality [8,9]. Allergies typically occur after the second exposure to the same or a closely related allergen. The first contact induces the production of IgE antibodies, leading to (hyper)sensitization to the respective venom, which then culminates in IgE-mediated allergic reactions upon subsequent contact [7].

However, distinct alternative medicine approaches utilize bee venom acupuncture (either using diluted apitoxin or actual bee stings), thus applying the whole blend of biologically active compounds as a treatment against cancer, immunological diseases such as rheumatoid arthritis and neurodegenerative disorders. Several deaths have been reported after these treatments [10,11], and a meta-analysis described a high risk for the occurrence of adverse events during therapy involving whole apitoxin [12]. The therapeutic potential of treatments with whole bee venom is rather disputed, and several meta-analyses have critically discussed the potential shortcomings of these studies [12,13,14]. Thus, the use of whole bee venom will very likely not gain wide acceptance in conventional medicine. Nevertheless, purified components of apitoxin, first and foremost melittin as the major constituent of bee venom [15], show interesting properties, which might become a therapeutic intervention strategy against several multifactorial diseases. In this review, we describe the biological properties of bee venom with a focus on melittin and discuss its therapeutic potential against cancer, neurodegenerative diseases and rheumatoid arthritis.

## 2. Melittin against Cancer

Cancer encompasses a heterogeneous group of diseases, defined by the uncontrolled division of cells and their potential for invading other tissues as a common phenotype. With an estimated number of 8.2 million deaths and 14.1 million new cases per year worldwide, cancer is a leading cause of death and an enormous global healthcare burden [16]. Despite the progress in cancer therapies achieved during the last decades, including surgery, chemotherapy, radiation or hormone ablation therapies, the most challenging problems such as recidivation, insufficient selectivity of toxic effects, or formation of resistance still remain, resulting in success rates lower than 50% [17]. Furthermore, access to a specialized treatment such as radiotherapy is limited, especially in low- and middle-income countries [18]. Thus, there is an urgent need for novel cancer therapeutics to overcome these problems, and one potential source for such substances might be bee venom.

Melittin, the major component of apitoxin, accounting for 40–60% of its total dry weight [3,5], is a 26 amino acid long amphipathic peptide [15]. Melittin induces cell death by disrupting biological membranes via the formation of pores and has haemolytic effects [15,19,20], suggesting a rather non-specific cytotoxicity of this peptide. Various approaches, including NMR spectroscopy, X-ray crystallography and electron microscopy, were used to characterize the mechanism of membrane lysis via melittin [4,21,22]. Melittin may translocate across biological membranes via transient pore fluctuations, resulting in stable pore formation if a distinct peptide/lipid ratio is given [21]. Although this lytic activity is described for all known melittin forms so far, the low abundant isoform melittin-S was suggested to have a lower haemolytic capacity compared to wild type melittin [23].

Despite this well-characterized general cytotoxicity of melittin, some studies have suggested this substance to specifically target cancer cells. One mode of action described in this respect involves Rac1, a Rho GTPase functioning in a wide range of physiological processes, including actin cytoskeleton regulation, axonal growth, adhesion, differentiation and mesenchymal-like migration. Rac1 hyperactivation is a common feature in various tumour cell types and is associated with increased metastatic ability [24]. Compared to normal liver cells or human hepatocellular carcinoma (HCC) cell lines with low metastatic potential, HCC cell lines with high metastatic potential display increased mRNA and protein levels of Rac1. Interestingly, increased Rac1 levels have been shown to sensitize these cells to the cytotoxic effects of melittin. Analysis of orthotopic transplanted tumour volume in mice revealed that melittin delayed tumour growth while increasing body weight [25]. As mice were sacrificed after 35 days of melittin treatment, the potential long-term effects of this substance cannot be evaluated. Although cell lines with high Rac1 levels were clearly sensitized towards melittin, high concentrations of this compound also triggered death of cells with low Rac1 levels (and thus low metastatic potential) upon prolonged incubation [25]. Thus, as for other anti-tumour drugs, the specificity of melittin towards cancer cells seems to be dose- and time-dependent and a general cytotoxicity of this apitoxin component cannot be disregarded.

Other studies have even described the selective toxicity of whole bee venom against cancer cells. The application of apitoxin was shown to have cytotoxic effects against different leukemic cells but not against normal bone marrow cells [26]. Cytotoxicity was determined using a 3-(4,5-dimethylthiazol-2-yl)-2,5-diphenyltetrazolium bromide (MTT)-assay, where the metabolic activity of a cell serves as a readout for cell viability. As cancer cells and healthy cells have markedly divergent metabolism, the use of additional, complementary methods, as suggested in [27], will be necessary to evaluate the tumour-specific cytotoxicity of apitoxin in these settings in more detail. In contrast to these findings, other studies have demonstrated a general toxicity of apitoxin during treatment of normal human lymphocytes compared to that during treatment of human lymphoma cell line HL-60 [28]. One explanation for this discrepancy might be that melittin, due to its positive charge, shows a slightly preferred binding to the surface of specific cancer cells that display an altered membrane lipid profile with increased negative surface charge compared to non-transformed cells, rendering it more potent against cancer cells, while healthy tissue is also affected [17,29].

A recent study evaluated the use of a hybrid peptide, composed of melittin and the cationic and amphipathic alpha-helix protein dKLA against tumour-associated M2-macrophages, the major component of tumour-infiltrating immune cells that promote tumour progression and contribute to chemotherapy-resistance [30]. dKLA is described to selectively disrupt mitochondrial membranes, thus inducing programmed cell death, while being unable to pass eukaryotic plasma membranes. When coupled to melittin, the authors suggest that this peptide can pass the membrane barrier, triggering apoptosis via cytochrome c release. Interestingly, this hybrid peptide decreased the portion of viable M2-macrophages to appr. 50%, while melittin alone showed appr. 75% viable cells after 24 hours of incubation. When sensitivity of M1 macrophages towards melittin-dKLA was tested, 66–86% were viable, thus other immune cells seem to be sensitive to this treatment as well. When treating subcutaneous tumour-bearing mice with both melittin and the hybrid peptide, the authors observed decreased tumour size and volume as well as decreased levels of tumour-associated macrophages compared to control mice. While this hybrid peptide represents a promising anti-cancer treatment, further analysis in respect to long-term effects and potential side effects of melittin-dKLA, are needed [30].

Despite the possibility of enhanced binding to cancer cells, a general and rather unspecific cytotoxicity of melittin is observed in most studies, limiting the potential for therapeutic approaches [15,19,31,32,33,34,35]. A recent study demonstrated rapid effects of melittin on gastric and colorectal cancer cells already over a period of 15 minutes, with first effects involving membrane damage, indicated by swelling, breakage or blebbing, observable within the first 30 seconds. Even though melittin indeed inhibited growth of both cancer cell types tested, the authors preclude administration of pure melittin in cancer therapy and instead suggest the use of melittin-conjugates and -derivates [36]. In aggregate, the general tenor of research suggests that a direct and unmodified application of bee venom or its components is not applicable for therapies at present, as clear specificity towards cancer cells is lacking.

However, chemical modifications of melittin and biotechnological approaches involving nanoparticle carriers might present a way to overcome certain limitations of melittin, taking advantage of the potential anticancer effects of this peptide. Melittin/avidin conjugates, which are cleavable by matrix metalloproteinase 2 (MMP2), were designed to directly target the cytotoxic effects of melittin to cancer cells [35]. MMP2 is overexpressed in several tumour variants, and treatment of prostate and ovarian cancer cells with melittin/avidin resulted in prominent cell death induction, monitored via lactate dehydrogenase release as well as live/dead fluorescence staining. Only minor toxicity of this melittin derivate was observed in healthy mouse fibroblasts with low MMP2 activity [35]. Another approach to circumvent the general cytotoxicity of melittin involves its insertion into nanoparticles. In general, the idea behind these nanoparticles is to stably incorporate a therapeutic substance into an inert carrier, which transports the substance to a certain location (e.g., specific tissue or cell type), where it is released and can mediate its function. Although the incorporation of melittin into liposomes failed due to inadvertent membrane disruption, perfluorocarbon nanoparticles showed a stable insertion of melittin [20]. Targeting of these nanoparticles was achieved by integrating αvβ3 integrin ligands [20], which bind to the respective αvβ3 integrins that are overexpressed in endothelial cells upon angiogenesis [37]. Integrating melittin into these nanoparticles resulted in reduced haemolysis, especially in concentrations below 10 µM. Furthermore, this study showed that these nanocarriers could be used to selectively deliver melittin to tumour cells, thereby reducing tumour growth in mice [20]. More recently, an environment-sensitive melittin delivery system was developed, combining zwitterionic glycol chitosan and disulfide bonds [38]. With this system, termed dual secured nano-sting, the haemolytic effects of melittin were nearly completely abolished, while cell death in different cancer cell lines was induced by an intracellular release of melittin, leading to mitochondrial damage [38]. The potential usability of this promising strategy in animals remains to be investigated.

In a recent study, melittin has been trapped inside nanoparticles and tested for its effects on liver sinusoidal endothelial cells (LSECS), the cells responsible for the immunologic tolerance of the liver and thus a common site for visceral metastases. Even though melittin *per se* was described to be specifically targeted to LSECS, application of this compound alone was again obstructed by haemolysis as its main side effect. Thus, the authors developed a 20 nm core shell peptide-lipid nanoparticle, where its lipid layer shielded melittin toxicity, making it useable for injection, while retaining melittin-induced toxicity in tumour cells. Intravenous administration of these nanoparticles led to a strong immunomodulation of LSECS and further blocked metastasis formation. Moreover, this treatment significantly prolonged survival rates in a spontaneous liver metastatic model of breast cancer, rendering these designed lipid-peptide hybrids one of the most promising therapeutics described so far [39].

These systems, as well as related approaches, might represent attractive strategies to harness melittin for cancer treatment, as this peptide harbours lytic activity, induces apoptotic cell death via inactivation of NF-κB [40], reduces liver cancer cell metastasis via inhibition of Rac1 [25] and impedes epidermal growth factor-induced breast cancer cell invasion [41].

Interestingly, the application of whole bee venom showed effects on cell viability and tumour cell migration comparable to melittin alone, while apamin, another bee venom peptide, had no effect on cell death and only slightly affected cell migration [41], suggesting that the observed anticancer effects of whole bee venom are likely induced by melittin. Nevertheless, apamin inhibits epithelial-mesenchymal transition in hepatocytes [42], a process observed during development, tumour progression and malignant transformation [43].

Together, these results suggest that some components of apitoxin, first and foremost melittin, exhibit interesting cytotoxic properties that could be used for therapeutic interventions against cancer. The rather unspecific cytotoxicity of melittin clearly limits this approach but might be overcome by using safe and directed targeting methods, including chemical modifications of melittin and incorporation into nanoparticles (Figure 1). Further research is needed to test these applications for anticancer effects, specificity and safety.

## 3. Apitoxin and Melittin in Neurodegenerative Diseases

In addition to its potential role against cancer, melittin has been suggested to inhibit neuroinflammatory processes associated with several neurodegenerative diseases. Neuroinflammation results from chronic activation of glial cells, such as astrocytes and microglia, and is a common feature in neurodegenerative disorders, including but not limited to Parkinson’s disease (PD) [44], Alzheimer’s disease (AD) [45] and amyotrophic lateral sclerosis (ALS) [46].

A histological hallmark of PD is the accumulation of intracellular protein deposits, so-called Lewy Bodies, in dopaminergic neurons of the *substantia nigra pars compacta*. α-Synuclein is a major component of these proteinaceous inclusions, and its aggregation and deposition are intimately linked to cell death of respective neurons, a major event in PD pathology [47]. Interestingly, the promoter sequence of tumour necrosis factor α (TNF-α) was hypomethylated in *substantia nigra pars compacta* samples of PD patients, resulting in increased promoter activity and elevated transcription of TNF-α [48]. This inflammatory cytokine can bind to two receptors, TNFR1 and TNFR2. Via a complex downstream network (e.g., reviewed in [49]), the binding of TNF-α to its receptors stimulates transcriptional responses, including activation of the transcription factor NF-κB. In parallel, TNF-α results in the release of IL-1α [50], which is also capable of activating NF-κB. This transcription factor is regarded as a “first responder” during inflammation and stimulates the expression of various cytokines. Furthermore, binding of TNF-α to TNFR1 results in the activation of a proteolytic cascade that is mediated and executed by caspases, subsequently leading to mitochondrial release of cytochrome c and apoptotic cell death.

TNF-α levels were also increased in *post mortem* brain samples of AD patients [51] and treatment of microglia with amyloid beta peptide (aβ), a major key player forming characteristic protein deposits in AD pathology, showed increased levels of TNF-α [52]. In addition to elevated TNF-α in PD and AD, this proinflammatory cytokine was induced in ALS patients [53] and in astrocytes expressing ALS-associated mutant forms of the fused in sarcoma (FUS) gene [54]. Overall, these results indicate an important role of neuroinflammatory processes involving TNF-α signalling in the pathogenesis of neurodegenerative diseases.

To investigate the potential effects of bee venom and melittin on these inflammatory responses in BV2 microglial cells, cultured microglia were treated with lipopolysaccharide (LPS) to induce inflammation. Compared to cells treated with LPS only, cells co-treated with bee venom or melittin showed reduced expression of TNF-α, IL-1β and IL-6 [55]. As cell viability seemed reduced, it remains to be investigated in more detail if the reduced levels of cytokines were not simply due to general cytotoxicity induced by bee venom/melittin-treatment [55]. In a mouse model of ALS based on the expression of mutated superoxide dismutase (SOD1), the subcutaneous injection of 0.1 µg/g melittin significantly improved motor function over a period of 15 days compared to control animals that received saline injections [56]. This beneficial effect was accompanied by an improvement in proteasomal function and a massive reduction in TNF-α levels in the spinal cord of melittin-treated animals compared to those in control animals. Nevertheless, Map2, as a marker for neuronal cells, was also reduced upon melittin treatment, which could indicate general neuronal cell death. [56]. Indeed, increased cell death has been detected upon treatment of primary rat neurons with melittin or with phospholipase A2, another component of bee venom [57]. The authors also performed studies with rats and observed changes in electroencephalograms (EEGs) for both components as well as neuropathological alterations in cortical, subcortical and forebrain neurons upon phospholipase A2 treatment [57]. In this line, neurotoxicity was also observed in magnocellular and parvocellular nuclei in the hypothalamus as well as in neurons of the *cerebral cortex* upon subcutaneous bee venom injection [58,59]. In this study, the authors tested different setups, including a low bee venom dose of 700 µg/kg (described as the amount released by one bee sting) in either a single treatment or in a subchronic treatment for 30 days [59]. Thereby, a single treatment seemed to be a potent neurostimulator, leading to epileptiform discharges without cellular lesions, whereas subchronic treatment resulted in EEGs similar to those of AD and PD patients, suggesting abnormal loss of neurons. This neuronal loss was confirmed with microscopic approaches, and electron microscopy analysis revealed phagocytic activity of glial cells and neurons with empty dendrites and axons, suggesting alterations in axoplasmic transport. In rats, a very high sublethal single dose of bee venom (62 mg/kg) triggered pronounced neurotoxic effects [59]. Thus, this study suggests that apitoxin mediates dosage-dependent neurotoxicity and that chronic application of bee venom at low dosages for an excessive period or with too short intervals between treatments can also cause neuronal cell death. However, the application of very low dosages of apitoxin with longer intervals resulted in neuroprotective effects in a PD mouse model, based on chronic 1-methyl-4-phenyl-1,2,3,6-tetrahydropyridine (MPTP)/probenecid treatment, a common regime to mimic PD pathology [60]. MPTP decreased the amount of hydroxylase-positive neurons in the *substantia nigra pars compacta*, whereas low amounts of bee venom (12 µg/kg or 120 µg/kg; based on dosages used in human desensitisation protocols applied for bee sting allergies) reduced the toxic consequences of MPTP [60]. Lower concentrations of bee venom with longer intervals between injections (3.5 days) were used in this work compared to those in the studies described above, and this milder approach indeed mediated neuroprotection in mice. Surprisingly, cytokines IL-1β and IL-6 were not increased upon MPTP treatment *per se*, and TNF-α levels were even reduced in this model, whereas the co-treatment of MPTP and bee venom enhanced IL-6 and TNF-α levels [60]. Since this chronic MPTP/probenecid treatment generally results in massively induced neuroinflammation, accompanied by increased levels of TNF-α [61], the observed reduction in cytokines seems counterintuitive. A recent study on potential cytoprotective effects of bee venom components suggests that phospholipase A2, but not melittin, protects against neuroinflammatory processes in a MPTP mouse model of PD [62]. One day after challenging mice with MPTP, phospholipase A2 and melittin were administrated for six consecutive days (0.5 mg/kg via subcutaneous injection) and interestingly, phospholipase A2-treated animals showed better motor function than the untreated controls, whereas no significant effect was observed for mice that received melittin injections. In addition, phospholipase A2-treatment inhibited loss of dopaminergic neurons in the *substantia nigra* in the MPTP-treated mice. The authors suggest that these effects were mediated by a stimulation of regulatory T cell differentiation, combined with inhibition of inflammatory Th1 and Th17 cell differentiation. As this study did not evaluate control animals without MPTP treatment that only received phospholipase A2 and melittin, further studies addressing the effects of these components on healthy individuals as well as potential long-term effects are needed [62].

Although potential neuroprotective effects of apitoxin might be highly relevant for future PD therapy strategies [60,62], a subsequent clinical trial could not confirm the beneficial effects of this treatment [63]. Within the last 15 years, several clinical studies with whole bee venom have been performed. One of these clinical trials investigated the potential effects of bee venom acupuncture as an adjuvant therapy in patients with idiopathic PD [64]. Patients were assigned into three groups, receiving either bee venom acupuncture (*n* = 13), acupuncture without bee venom (*n* = 13) or no treatment (*n* = 9) over a rather short period of 8 weeks. The analysis of the Unified Parkinson’s Disease Rating Scale (UPDRS), aiming to describe the longitudinal course of PD [65], showed a significant improvement for both bee venom acupuncture and acupuncture groups compared to that for the control group [64]. Thus, these effects cannot be attributed to bee venom. In a second study, the authors compared 11 PD patients before and after adjuvant treatment with bee venom [66]. Although improvement in the UPDRS due to bee venom acupuncture has been observed, it remains unclear whether the beneficial effects were due to apitoxin, as acupuncture without bee venom was not included in this trial. In contrast to these findings, a recent double-blinded and placebo-controlled trial did not detect clear symptomatic or disease-modifying effects of bee venom injections in PD patients [63]. Subcutaneous injections of 100 µg bee venom in 1 mL NaCl 0.9% (a setup used for immunotherapy to treat bee sting allergy) were compared to injections of placebo treatment (1 mL NaCl 0.9%) over a course of 11 months (*n* = 20 in each group). No significant differences between the groups were observed in the UPDRS or in other tested scores. The total PDQ-39 score (a PD-specific health status questionnaire) was also not significantly different between both groups, while the “activities of daily living” as a subsection of the PDQ-39 was even worse in the bee venom group than in the placebo group [63].

In addition to these clinical trials with PD patients, another study aimed to investigate the therapeutic potential of bee venom in multiple sclerosis patients [67]. Over a period of 24 weeks, live bees were administered three times a week with a continuously increasing number of bee stings (one additional sting per session) with up to a maximum number of 20 bee stings per treatment. After 24 weeks, the treatment group switched to no treatment, and the control group started receiving bee stings for a period of 24 weeks (crossover study design; *n* = 12/13 analysed). The authors investigated MRI brain images for changes in lesions, relapse rate, disability, fatigue and health-related quality of life and could not find any changes mediated by bee venom therapy [67]. The authors concluded that multiple sclerosis patients should be advised to refrain from bee venom therapy unless evidence for a beneficial effect is presented.

In sum, controversial findings exist, describing either a neuroprotective or a neurotoxic consequence of bee venom therapy or application of its components in preclinical models for neurodegenerative diseases (summarized in Figure 2). Although some clinical trials suggest protective effects of bee venom acupuncture against PD, others describe no effect on PD or multiple sclerosis. Given the cytotoxic effects of melittin and other components of apitoxin, the high prevalence and severity of bee venom allergies [6], the massive risk of apitherapies [12] and currently missing evidence for beneficial effects in patients with neurodegenerative disorders, the application of whole bee venom as a therapy against neurodegenerative diseases will most likely find no usability in conventional medicine in the near future. However, to the best of our knowledge, no clinical trials with distinct bee venom components, such as melittin, have been conducted thus far. Further investigation of the single components of apitoxin (e.g., phospholipase A2, as described above) as well as of chemically/biotechnologically modified forms of these substances are needed to evaluate their potential to counteract neurodegenerative diseases.

## 4. Apitoxin and Melittin against Rheumatoid Arthritis

Rheumatoid arthritis (RA) belongs to the most common inflammatory arthropathies and is a chronic autoimmune disorder, leading to pain and stiffness, swelling and deformity of joints. Clinical disease onset is preceded by a pre-RA phase lasting up to several years that is characterized by circulating autoantibodies, elevated concentrations of inflammatory chemokines and cytokines (e.g., TNFα, IL-1) as well as a changed metabolism. Aggressive treatment strategies and immune therapy help to slow down disease progression, but no cure exists at present [68]. As some RA patients do not respond to conventional treatment strategies, the development of new therapies is essential [69]. In this line, bee venom and especially its component melittin have come into focus, with several studies ascribing anti-inflammatory effects that might be exploited for novel RA treatment strategies.

The most commonly used animal model to investigate RA pathology and examine potential agents usable for treatment strategies is the collagen-induced arthritis (CIA) mouse/rat model. To provoke autoimmune arthritis, animals are immunized with an emulsion consisting of collagen type II and complete Freud’s adjuvant, leading to a disease progression that is highly reminiscent of RA, including synovial hyperplasia, mononuclear cell infiltration, and cartilage degradation [70]. Bee venom injected in a certain acupoint was shown to have anti-nociceptive effects on a rat model of CIA, while saline treatment as well as non-acupoint apitoxin treatment was without effect. The peak of reduced nociception, evaluated by tail flick latency, was reached 30 minutes after the treatment and lasted for approximately 1 hour. Interestingly, although this effect did not change in the presence of an µ-opioid receptor antagonist, pre-treatment with an α_2_-adrenergic receptor antagonist caused significant elevated nociception, leading to the assumption that the observed anti-nociceptive effects seemed to rely on α_2_-adrenergic pathways [71]. In contrast, another study found that melittin induced local inflammation, ongoing pain and hypersensitivity when injected into rats, provoking the longest and most intense nociceptive responses compared to those induced by other apitoxin components [72]. Furthermore, apitoxin was suggested to inhibit cytokine production (e.g., IFN-γ, IL-1β, TNFα) in a CIA rat model at higher doses and further slowed down disease progression in a dose-dependent manner [73]. The water-soluble fraction of apitoxin (which contained, among other components, melittin) was further demonstrated to suppress paw oedema and radiological changes in a CIA rat model and to reduce IL-6 levels and nociceptive behaviour. In this study, the authors suggest that not only one substance alone but the complex mixture of compounds present in the water soluble apitoxin fraction mediated the observed positive effects [74]. Bee venom was also shown to reduce the levels of several cytoplasmic, lysosomal and extracellular proteases, but the mechanism behind this as well as its relevance remains to be explored [75].

Several studies analysed the potential positive effects of apitoxin/melittin on RA in cell culture using either synoviocytes extracted from patients or different immortalized cell lines (e.g., mouse macrophages). A study conducted with a mouse macrophage cell line and synovial tissue from RA patients observed the suppression of IκKα and IκKβ activity by both melittin and bee venom [76]. At present, inhibitors of IκK are used for the treatment of inflammatory diseases since they block IκB release, thus preventing NF-κB activity. The authors further demonstrated that the inhibitory effect observed for bee venom was most likely due to an interaction of melittin with cysteines in the active domains of both IκKα and IκKβ, and mutation of these cysteines to alanines abrogated the effects. Melittin used at concentrations of 10 µg/mL showed comparable results as whole bee venom and further reduced inflammation provoked by LPS [76]. In addition, apitoxin and melittin were capable of reducing NF-κB activation by LPS, most likely via direct interaction with the p50 subunit of NF-κB [77]. Although these melittin-provoked effects are clearly promising for RA treatment, further studies and in-depth analysis of potential negative side effects such as cytotoxicity are needed [76,77]. In contrast to the results discussed above, another study did not find any blockade of NF-κB activation or alterations in IκB in response to either bee venom or melittin [78]. Instead, a moderate upregulation of the mRNA levels of proinflammatory genes was observed upon treatment. Furthermore, bee venom in quantities higher than 10 µg/mL was described to result in membrane disintegration of all cells tested. Interestingly, in contrast to anti-inflammatory reagents that block the MAP kinase (MAPK) pathway, treatment with apitoxin or melittin triggered the phosphorylation and subsequent activation of MAPK [78], further challenging the proposed anti-inflammatory impact on RA [76,77]. An additional feature of RA is the resistance of synoviocytes to apoptosis. Melittin has been shown to trigger apoptosis in apoptosis-resistant synoviocytes from patients with advanced RA via downregulation of IL-6-induced NF-κB activation, suppression of anti-apoptotic genes and a concomitant increase in the expression levels of pro-apoptotic factors and caspase activity [79].

In summary, several studies have demonstrated the anti-inflammatory effects of melittin and bee venom in general, highlighting the potential of these natural compounds to mitigate RA symptoms (summarized in Figure 3). Nevertheless, controversial findings exist, and further analyses of long-term effects and specifically general cytotoxicity will be essential to assess applicability.

## 5. Conclusions

Natural products are an important and valuable source for the identification and development of novel drugs. Among natural sources, diverse components of toxins from scorpions, snakes or bees are handled as promising therapeutic tools to treat neurodegenerative or inflammatory diseases. Bee venom and its components may ameliorate pathologies associated with neurodegenerative disorders such as PD, AD and ALS, and accumulating evidence hints at their neuroprotective and anti-inflammatory activity. Nevertheless, controversial results exist at both preclinical and clinical levels, complicating the precise evaluation of potential beneficial effects. As bee venom is a mixture of various biologically active substances, including melittin as a toxic peptide that results in the lysis of biological membranes [21], the exploration of isolated single components rather than of the complex mixture might provide more consistent results. Distinct components of bee venom might induce unexpected side effects. Along this line, a recent study provided evidence for interference of subcutaneously injected phospholipase A2 and melittin with spermatogenesis in rats [80]. The use of whole bee venom for therapeutic application might be questionable, not only due to inconsistent beneficial effects but also because of potential side effects and the high risk of this bee venom therapy [12]. Thus, further studies should focus on single components of apitoxin and modified variants of these molecules to investigate potential effects against neurodegenerative diseases as well as RA, with a critical evaluation of toxic side effects.

To date, the most promising therapeutic usage of apitoxin components might be the application of melittin against cancer. Although the results regarding a specific anticancer function of melittin seem rather inconsistent, and unspecific cytotoxicity of this peptide has often been reported, the direct targeting of modified melittin to cancer cells (e.g., with nanoparticles), resulting in the selective uptake and lysis of transformed cells [38], might present an attractive approach. Future research evaluating specificity and safety of these applications will show, whether targeted melittin-delivery might be indeed usable in cancer therapy.

## Figures and Tables

**Figure 1 toxins-12-00066-f001:**
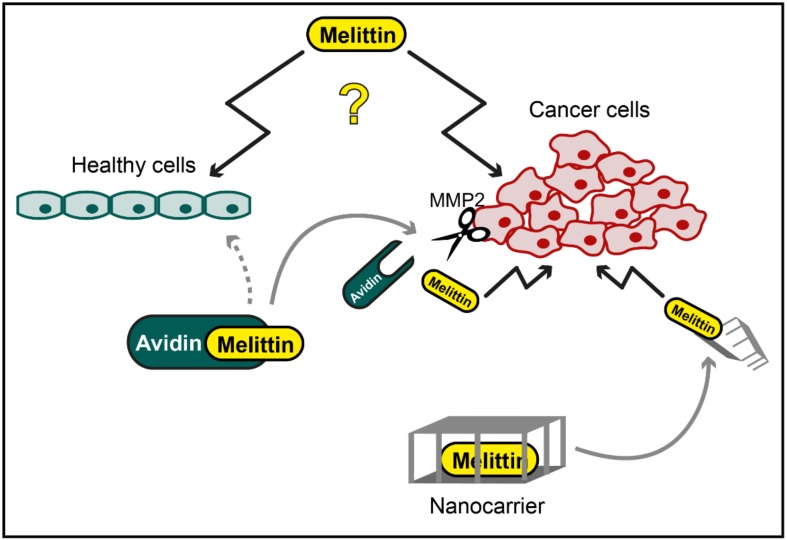
Potential application of melittin in cancer therapy. While specific cytotoxic effects of melittin against cancer cells have been reported, melittin has also been shown to mediate unspecific toxicity towards healthy cells (illustrated as question mark). However, chemical modification, e.g., by conjugation of melittin to avidin, which is cleaved and activated by cancer cell-specific matrix metalloproteinase 2 (MMP2), or cancer cell-directed targeting of melittin with nanocarriers might be used to overcome this unspecific toxicity.

**Figure 2 toxins-12-00066-f002:**
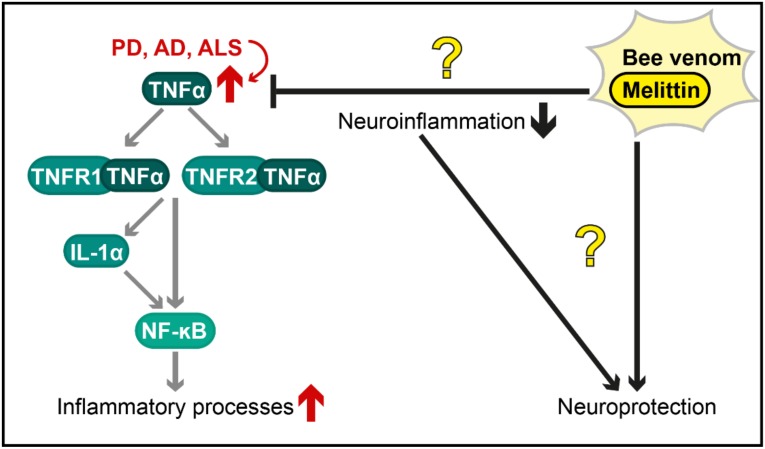
Postulated therapeutic effects of bee venom and melittin against neuroinflammation in diverse neurodegenerative diseases. The tumour necrosis factor α (TNF-α) pathway (illustrated in a simplified way) is an important mechanism by which inflammatory processes are activated and is overstimulated in various neurodegenerative disorders (red arrows), including Parkinson’s disease (PD), Alzheimer’s disease (AD) and amyotrophic lateral sclerosis (ALS). The protective effects of bee venom and melittin (shown with black arrows for activation and black bar-headed arrows for repression) are controversial (illustrated as question marks in the figure). Further research is required to characterize these effects and to evaluate, whether they are caused by a reduction in neuroinflammation or by other mechanisms. TNFR1/2 = Tumour necrosis factor receptor 1/2; IL-1α = Interleukin 1α; NF-κB = Nuclear factor kappa-light chain enhancer of activated B cells.

**Figure 3 toxins-12-00066-f003:**
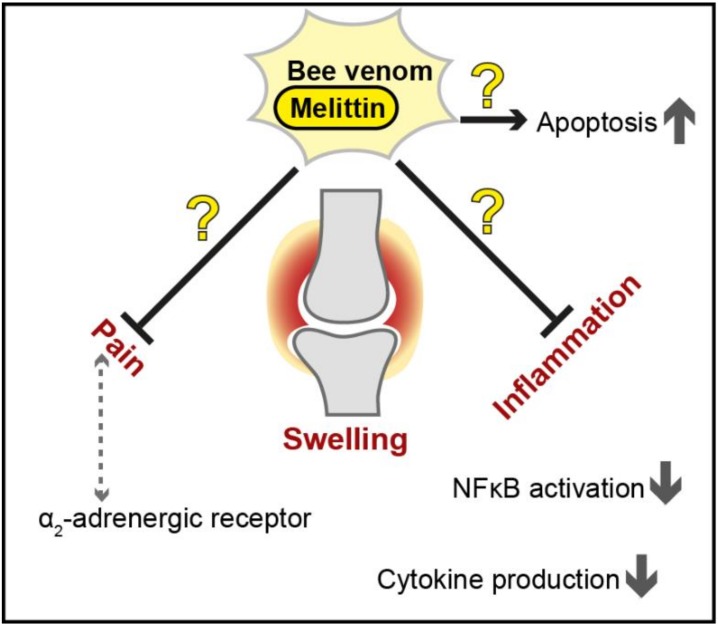
Proposed therapeutic effects of bee venom and melittin against inflammation and pain in rheumatoid arthritis. The protective effects of bee venom and melittin (shown as black arrows for activation and black bar-headed arrows for repression) are controversial (illustrated as question marks in the figure), and further investigation is needed to assess the usability of apitoxin and/or melittin as a novel treatment strategy in this multifactorial disease. NF-κB = Nuclear factor kappa light chain enhancer of activated B cells.

**Table 1 toxins-12-00066-t001:** Important components of bee venom (adapted from [3,5]).

Class of Molecule	Apitoxin Component	Percent in Dry Venom
Small proteins and peptides	Melittin	40–60
	Apamin	1–3
	Mast cell degranulating peptide	1–3
	Adolapin	0.1–1
	Tertiapin	0.1
	Cardiopep	0.7
	Procamine A, B	1–2
	Secapine	0.5–2
	Minimine	2–3
	Pamine	1–3
Enzymes	Phospholipase A2	10–12
	Phospholipase B	1
	Acid phosphomonoesterase	1
	Hyaluronidase	1–3
	Lysophospholipase	1
	α-Glucosidase	0.6
Amines	Histamine	0.5–2
	Dopamine	0.13–1
	Noradrenalin	0.1–0.7
Sugars	Glucose, fructose	2–4
Minerals	Phosphate, calcium, magnesium	3–4

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
