# Peer review of "Apitoxin and Its Components against Cancer, Neurodegeneration and Rheumatoid Arthritis: Limitations and Possibilities"

_toxins, 2020, doi:10.3390/toxins12020066_

Round 1
Reviewer 1 Report
This is an extremely well written review. I only detected one mistake on line 260 where "individuum" should bee "individual." While some of the topics covered in the present review have been reviewed before in Toxins (Hwang et al 2015), the current work provides a nice update to these topics. Additionally, the review extends the coverage to the potential of bee venom/mellitin as a cancer therapy.
I think this review is a good addition to the comprehensive and timely reviews in Toxins.
Author Response
Point 1: This is an extremely well written review. I only detected one mistake on line 260 where "individuum" should bee "individual." While some of the topics covered in the present review have been reviewed before in Toxins (Hwang et al 2015), the current work provides a nice update to these topics. Additionally, the review extends the coverage to the potential of bee venom/mellitin as a cancer therapy.
Response 1: We thank the referee for these positive comments and corrected this mistake.
Reviewer 2 Report
The review addresses a very important issue (use of bee venom), which has been extensively studied in the last 10 years.
However, the title of the manuscript is the application of melittin and apitoxin, and several paragraphs mention the effect of the whole bee venom, as well as the activity of other toxins (phospholipases A2, for example). Moreover, there is an item about the components of bee venom, what emphasizes the bee venom contribution. Thus, in my opinion, the authors should adequate the title in order to fit it to the manuscript, and in this sense, the information about other toxins should not be removed, as is relevant as well.
The keywords should be reviewed, once the use of Parkinsons´s Disease is a very specific term, as all neurodegenerative diseases are been considered.
Some references should be reviewed. For example, the article of Oršolić is not the best one to provide information about the bee venom composition, please verify. Moreover, there is a work that demonstrated a melittin isoform (termed melittin-S) with less lytic effect and could be cited when the lytic mechanism of melittin is mentioned. Besides, references for the table 1 should be added.
In the line 40 it is mentioned the allergic potential of bee venom. This characteristic should be explored in the manuscript, as is an important issue to be considered in the venom use for therapy.
The legend in the figure 1 mention that the antitumor effect of melittin is controversially discussed, but there are several articles showing a clear tumor reduction (one of them mentioned in the manuscript) and the mechanism of action (some discussed here). Thus, although unspecific, the cytotoxic activity is a fact and the controversial statement should be removed. Moreover, the discussion should be detailed, as there are much more works about it that are not mentioned, including other bee venom components, such as phospholipases A2.
Line 207, rewrite in order to have a smaller paragraph.
Regarding rheumatoid arthritis and pain, there are several works that demonstrate the nociceptive effect of melittin, and according to the authors, only one works describe as antinociceptive effect. Is the antinociceptive a consistent data to be included in a review?
Author Response
We thank the referee for his/her constructive comments. All suggestions have been addressed, and the changes in the manuscript text are highlighted in yellow.
Point 1: The review addresses a very important issue (use of bee venom), which has been extensively studied in the last 10 years. However, the title of the manuscript is the application of melittin and apitoxin, and several paragraphs mention the effect of the whole bee venom, as well as the activity of other toxins (phospholipases A2, for example). Moreover, there is an item about the components of bee venom, what emphasizes the bee venom contribution. Thus, in my opinion, the authors should adequate the title in order to fit it to the manuscript, and in this sense, the information about other toxins should not be removed, as is relevant as well.
Response 1: The title has now been adapted to state more clearly that this review is about whole apitoxin and several of its components (not only melittin). The title now reads as follows: “Apitoxin and its Components against Cancer, Neurodegeneration and Rheumatoid Arthritis: Limitations and Possibilities”.
Point 2: The keywords should be reviewed, once the use of Parkinson´s Disease is a very specific term, as all neurodegenerative diseases are been considered.
Response 2:We agree and removed Parkinson’s disease as keyword.
Point 3: Some references should be reviewed. For example, the article of Oršolić is not the best one to provide information about the bee venom composition, please verify. Moreover, there is a work that demonstrated a melittin isoform (termed melittin-S) with less lytic effect and could be cited when the lytic mechanism of melittin is mentioned. Besides, references for the table 1 should be added.
Response 3: We thank the referee for this critical comment. We re-evaluated all given values in table 1 with recent research, among others a very recent review article published in “Frontiers in Immunology” by Pucca et al. (2019), and could not find any significant discrepancies. Where applicable, we made slight changes. In addition, we added the paper by Pucca et al. as an additional reference. Please note that the references are placed in the figure heading.
A respective sentence for the description of melittin-S with an adequate reference was added (lines 102-104)
Point 4: In the line 40 it is mentioned the allergic potential of bee venom. This characteristic should be explored in the manuscript, as is an important issue to be considered in the venom use for therapy.
Response 4: We agree with the referee and have now included an additional section that provides some information about allergens in bee venom, different allergic reactions and the IgE response (please see page 3, lines 61-69)
Point 5: The legend in the figure 1 mention that the antitumor effect of melittin is controversially discussed, but there are several articles showing a clear tumor reduction (one of them mentioned in the manuscript) and the mechanism of action (some discussed here). Thus, although unspecific, the cytotoxic activity is a fact and the controversial statement should be removed. Moreover, the discussion should be detailed, as there are much more works about it that are not mentioned, including other bee venom components, such as phospholipases A2.
Response 5: Our review article aims to critically explore potential therapeutic applications of apitoxin and its components, like melittin. We of course completely agree that the cytotoxic activity of melittin has been proven. However, while some studies show a clear tumor reduction and anti-cancer effects of melittin, others also have shown that melittin can exert cytotoxicity towards healthy cells, which might be due to the unspecific lytic potential of melittin. We also extensively discuss controversial findings regarding phospholipase A2. Highlighting these controversies is one of the aims of this review and in our view important for future research. Hence, we would prefer to keep this statement in the figure legend. Still, we slightly re-phrased to avoid the use of “controversially discussed”, and the modified sentence now reads as follows: “While specific cytotoxic effects of melittin against cancer cells have been reported, melittin has also been shown to mediate unspecific toxicity towards healthy cells”.
Point 6: Line 207, rewrite in order to have a smaller paragraph.
Response 6: The effects of apitoxin and its components against (neuro)inflammation and neurodegeneration are an important chapter in our manuscript, and we tried to give an overview in a very comprehensive way. To be able to discuss details about the proposed modes of action of bee venom components against different neurodegenerative diseases, we think that it is important to first provide a brief background of the relevant information for these neurodegenerative diseases. At present, the introduction into some aspects of Parkinson’s disease, Alzheimer’s disease, and amyotrophic lateral sclerosis as well as associated neuroinflammation occupies about half a page and cannot be shortened without losing important information.
Point 7: Regarding rheumatoid arthritis and pain, there are several works that demonstrate the nociceptive effect of melittin, and according to the authors, only one works describe as antinociceptive effect. Is the antinociceptive a consistent data to be included in a review?
Response 7: We agree with the referee and discuss several works on nociceptive effects of apitoxin and melittin (please see lines 371-385 for the paragraph dedicated to nociceptive and anti-nociceptive effects). Indeed, we only found two studies describing anti-nociceptive effects (one for apitoxin, one for melittin-containing fractions), which are both cited in our manuscript. We completely agree with the referee that there is a higher number of studies showing nociceptive effects. Still, as we aim for an unbiased elaboration of the literature, we also critically discussed the data mentioned by the referee.
Reviewer 3 Report
The authors discuss therapeutic potential of bee venom components on the diverse human diseases including cancer, neurodegenerative disorders and rheumatoid arthritis. They describe current knowledge of therapeutic effects and mechanisms of bee venom components.
Overall, this is an interesting and timely review. The topic is interesting in the field of development of alternative approaches to human diseases. The paper is in general balanced and comprehensive. I have only a few suggestions to strengthen the flow and impact or this review.
In the section 2 (melittin against cancer), the authors described the cytotoxicity of bee venom component melittin. The discrepancy between selective toxicity of whole bee venom against cancer cells and a general unspecific toxicity of bee venom component was described in the manuscript. The authors also discussed the recent approaches to enhance anticancer effect and specificity, such as using avidin-conjugated or nanoparticle-trapped melittin. In terms of avidin/melittin conjugate, the specific effect against cancer cells via MMP2-mediated cleavage was well-described. However in the case of other methods such as nanoparticle-trapped melittin (in line 139-155 and figure 1), how this method could enhance specificity of melittin on cancer cells was not described. Please elaborate 1 or 2 lines to enable a more clear understanding by the readers.
The therapeutic effect of bee venom and melittin on neurodegenerative diseases was described in section 3, with the major focus on the cytokine modulation and PD. The inconsistencies of effectiveness were described, and the authors wrote that “ no clinical trials with distinct bee venom components, such as melittin, have been conducted thus far” (line 301). I recommend the authors to suggest potential candidates (e.g. phospholipase A2) for the alternative treatment based on recent studies.
In the section 4 and figure 3, the therapeutic effects of bee venom and melittin against inflammation and pain in rheumatoid arthritis were described. I agree with the authors’ view that the mechanisms of analgesic effect and anti-inflammatory effect would not be same. In the current manuscript the concepts of the bee venom-mediated therapeutic effects on the RA are somewhat vague, and the pathway mediating swelling described in figure 3 was not covered in the manuscript text. In my opinion the mechanisms involved in swelling in not that important and not independent to inflammation mechanisms. I recommend the authors to refine the section 4 and figure 3 for clear understanding of the readers.
Author Response
We thank the referee for his/her constructive comments. All suggestions have been addressed, and the changes in the manuscript text are highlighted in yellow.
The authors discuss therapeutic potential of bee venom components on the diverse human diseases including cancer, neurodegenerative disorders and rheumatoid arthritis. They describe current knowledge of therapeutic effects and mechanisms of bee venom components.Overall, this is an interesting and timely review. The topic is interesting in the field of development of alternative approaches to human diseases. The paper is in general balanced and comprehensive. I have only a few suggestions to strengthen the flow and impact or this review.
Point 1: In the section 2 (melittin against cancer), the authors described the cytotoxicity of bee venom component melittin. The discrepancy between selective toxicity of whole bee venom against cancer cells and a general unspecific toxicity of bee venom component was described in the manuscript. The authors also discussed the recent approaches to enhance anticancer effect and specificity, such as using avidin-conjugated or nanoparticle-trapped melittin. In terms of avidin/melittin conjugate, the specific effect against cancer cells via MMP2-mediated cleavage was well-described. However in the case of other methods such as nanoparticle-trapped melittin (in line 139-155 and figure 1), how this method could enhance specificity of melittin on cancer cells was not described. Please elaborate 1 or 2 lines to enable a more clear understanding by the readers.
Response 1: We agree with the referee and added a respective section in this paragraph to explain the general principle of these nanocarriers. Further, we also included a brief explanation in respect to the mode of action of the melittin-loaded perfluorocarbon nanoparticles used in the study of Soman et al., in particular how they are targeted to tumor tissue (lines 168-175)
Point 2: The therapeutic effect of bee venom and melittin on neurodegenerative diseases was described in section 3, with the major focus on the cytokine modulation and PD. The inconsistencies of effectiveness were described, and the authors wrote that “ no clinical trials with distinct bee venom components, such as melittin, have been conducted thus far” (line 301). I recommend the authors to suggest potential candidates (e.g. phospholipase A2) for the alternative treatment based on recent studies.
Response 2: We agree that one of these potential candidates might be phospholipase A2. As pre-clinical studies about phospholipase A2 are already described in earlier sections of the same paragraph, we decided to just briefly state phospholipase A2 as one candidate as suggested by the referee, and referred to respective sections in our manuscript.
Point 3: In the section 4 and figure 3, the therapeutic effects of bee venom and melittin against inflammation and pain in rheumatoid arthritis were described. I agree with the authors’ view that the mechanisms of analgesic effect and anti-inflammatory effect would not be same. In the current manuscript the concepts of the bee venom-mediated therapeutic effects on the RA are somewhat vague, and the pathway mediating swelling described in figure 3 was not covered in the manuscript text. In my opinion the mechanisms involved in swelling in not that important and not independent to inflammation mechanisms. I recommend the authors to refine the section 4 and figure 3 for clear understanding of the readers.
Response 3: We thank the referee for this critical comment. Indeed, the mechanisms involved in swelling might not be that important and in addition not independent to the mechanisms associated with inflammation. We thus removed this from figure 3 and refined respective sections in the figure legend.